# Influence of the Polymeric Matrix on the Optical and Electrical Properties of Copper Porphine-Based Semiconductor Hybrid Films

**DOI:** 10.3390/polym15143125

**Published:** 2023-07-22

**Authors:** Maria Elena Sánchez Vergara, Joaquín André Hernández Méndez, Daniela González Verdugo, Isabella María Giammattei Funes, Octavio Lozada Flores

**Affiliations:** 1Facultad de Ingeniería, Universidad Anáhuac México, Avenida Universidad Anáhuac 46, Col. Lomas Anáhuac, Mexico City 52786, Mexico; joaquin.hernandez16@anahuac.mx (J.A.H.M.); daniela.gonzalezve@anahuac.mx (D.G.V.); 2Facultad de Ingeniería, Universidad Panamericana, Augusto Rodin 498, Mexico City 03920, Mexico; 0232941@up.edu.mx (I.M.G.F.); olozada@up.edu.mx (O.L.F.)

**Keywords:** polymethylmethacrylate, polystyrene, porphine, hybrid film, optical properties, electrical behavior

## Abstract

In this study, we assessed the electrical and optical behavior of semiconductor hybrid films fabricated from octaethyl-21H,23H-porphine copper (CuP), embedded in polymethylmethacrylate (PMMA), and polystyrene (PS). The hybrid films were characterized structurally and morphologically using infrared spectroscopy (IR), atomic force microscopy (AFM), scanning electron microscopy (SEM), and X-ray diffraction (XRD). Subsequently, the PMMA:CuP and PS:CuP hybrid films were evaluated optically by UV–vis spectroscopy, as well as electrically, with the four-point collinear method. Hybrid films present a homogeneous and low roughness morphology. In addition, the PS matrix allows the crystallization of the porphin, while PMMA promotes the amorphous structure in CuP. The polymeric matrix also affects the optical behavior of the films, since the smallest optical gap (2.16 eV) and onset gap (1.89 eV), and the highest transparency are obtained in the film with a PMMA matrix. Finally, the electrical behavior in hybrid films is also affected by the matrix: the largest amount of current carried is approximately 0.01 A for the PS:CuP film, and 0.0015 A for the PMMA:CuP film. Thanks to the above properties, hybrid films are promising candidates for use in optoelectronic devices.

## 1. Introduction

Organic semiconductors are structures that contain several carbon–carbon bonds and can carry out electronic conduction as inorganic semiconductors [1]. This type of semiconductor has different features such as low weight, transparency, higher brightness, flexibility, and ease to process, and can be used to fabricate inexpensive devices [2,3,4,5]. Some applications for these materials are electronic paper, supercapacitors, organic light-emitting diodes (OLEDs), thin-film batteries, organic photovoltaics, organic field-effect transistors, solar cells, sensors, and biosensors [2,3,4,5,6]. If the molecular size of organic semiconductors is taken into account, two types are distinguished: conductive polymers and small molecules. From a chemical point of view, there are significant differences between these categories, which directly affects their technological performance. Polymeric macromolecules are formed through the repetition of a main unit, which is the monomer. From its processing, the high molecular weight of the polymers favors the formation of thin films by solution, which benefits different applications such as its use in organic electronics [7]. On the other hand, small molecules have a conjugated structure, consisting mainly of aromatic or heteroaromatic rings. Thanks to their low molecular weight, they can be deposited as thin films by vacuum-thermal evaporation; however, they have the advantage that they also admit the alternative of being processed in a solution [7].

Porphines belong to the small molecule group and are identified as the core macrocycle of synthetic and natural porphyrins, also it is known as the simplest porphyrin [8]. It has a large π-conjugated system, high thermal stability, good photosensitizing ability [9], and high molar absorptivity at the Soret peak [10]. The use of porphines currently faces some challenges including that they are unfeasible to synthesize on a large scale and have high insolubility [8]. To find an answer to the solubility problem, porphines can be processed in the solid state. In organic electronics, porphyrin chemistry represents an extensive area of interest and has already reached the stage of advanced applications, where films of these materials are investigated for their use as chemical sensors for volatile organic compounds, information storage, and non-linear optical materials [11]. A starting point for the fabrication of highly functional thin films is the monolayer and multilayer formation. The use of covalently bound self-assembly is an important advantage because the weak van der Waals interactions or hydrogen bonding between interfaces are replaced with covalent bonds. One of the porphyrins that is considered the most helpful and has been utilized in the development of thin films is the manganese(III) porphyrin. Susuki et al. accounted for the manganese porphyrin as an active material [12], and Khorasani et al. as ionophores in thiocyanate-selective sensors [13]. Even so, the best result reported for film sensor was obtained with tetratolylporphine manganese (III) chloride (MnTTPCI); it has steady and reproducible electrochromic properties and possesses properties that serve to function as a detecting material for optical gas [14]. Moreover, functionalized porphyrins can be used in thin films such as sensitizers, in their free-base form, or with a redox inactive metal such as zinc or copper inside the macrocycle cavity. The above is because the porphyrins reveal strong light absorption with a Soret band in the 400–450 nm range and visible bands in the 500–650 nm range [15]. Due to this optical property, the highest efficiency organic solar cell reported is composed of Co(II/III)tris(bipyridyl)-based electrolyte and a donor-π-bridge-acceptor zinc porphyrin as a sensitizer (YD2-o-C8) [16].

Based on the properties and applications of this small molecule, this study proposes (i) the manufacturing of semiconductor hybrid films using copper porphine (CuP) in addition to polymethylmethacrylate (PMMA) and polystyrene (PS) as a matrix, (ii) the structural and surface topographic characterization of these hybrid films and (iii) the evaluation of their optical and electrical properties. The novelty of this work is related to the development of new hybrid films polymer-CuP that can be used in an emerging category of optoelectronic devices. In addition, a comparative study between the effects on the optical and electrical properties of PMMA and PS matrices was carried out. PMMA and PS are a category of non-conductive polymers that are used in molecular electronics, for the manufacturing of different types of devices such as: floating-gate memory [17,18], electret memory [17,19], optical data storage devices [7,20] and phototransistors [7]. Specifically, PMMA, as a polymeric waveguide has been highlighted for use in optoelectronic devices and as optical components because of its volume productivity and low cost [21]. In the case of PS it is considered one of the oldest commercial polymers used in molecular electronics as a dielectric layer or in composite films with fullerene derivatives or gold nanoparticles [22]. However, both polymers are also brittle materials and have poor chemical resistance, low melting points, and low flexibility. To solve the issues mentioned above, its properties can be modified [21,23]. These modifications include forming hybrid films with the polymer and some small molecules with semiconductor behavior. It is for this reason that this study focuses on the manufacturing of hybrid films that have the semiconductor characteristics of porphine, but with the possibility that these properties can be modulated, depending on the type of polymer matrix used.

## 2. Materials and Methods

Amorphous atactics polystyrene (PS; [CH_2_CH(C_6_H_5_)]_n_) and poly(methyl methacrylate) (PMMA; [CH_2_C(CH_3_)(CO_2_CH_3_)]_n_), and the 2,3,7,8,12,13,17,18-octaethyl-21H,23H-porphine copper(II) (CuP; C_36_H_44_CuN_4_) (see Figure 1) were obtained from commercial suppliers (Sigma-Aldrich, Saint Louis, MO, USA), and used without further purification. The hybrid films were deposited on Corning glass, n-type silicon, and fluorine-doped tin oxide-coated glass slide (FTO). It is important to emphasize that the glass and FTO substrates were previously washed in an ultrasonic bath with dichloromethane, methanol, and acetone. The n-silicon was washed with a “p” solution (10 mL HF, 15 mL HNO_3_, and 300 mL H_2_O). The hybrid films were fabricated from a dispersion with 6 mL of the polymer and CuP from a dilution of 10 wt% in toluene for PS and acetone for PMMA. The mixture was dispersed using the G560 shaker of Scientific Industries Vortex-Genie (Bohemia, NY, USA). A syringe was used to apply 0.6 mL on the surface and then spread the dispersion. Directly after spreading the dispersion, the films were brought to 55 °C for 5 min in the drying oven Briteg SC-92898 (Instrumentos Científicos, S.A de C.V.). This accelerated the hybrid film fabrication process and prevented the samples from swelling and cracking during drying. For the deposit of the pristine CuP film, the high vacuum evaporation technique was used, with a system (Intercovamex, S.A. de C.V., Cuernavaca, Morelos, México) of two pumps: mechanical and turbomolecular that evaporated the CuP at a vacuum of 10^−5^ torr, with a deposition rate of 4.6Å/s and until a thickness of 15 Å. The thickness was monitored using a microbalance quartz crystal monitor, connected to a thickness sensor.

The n-silicon substrates were used for infrared (IR) spectroscopy evaluation using the Nicolet iS5-FT spectrometer (Thermo Fisher Scientific Inc., Waltham, MA, USA) at a wavelength range of 4000 to 400 cm^−1^. Topographical characteristics were investigated on n-silicon substrates with a Nano AFM atomic force microscope, using an Ntegra platform, for this, the AFM was operated in static force contact mode, using the Stat0.2LauD silicon tip (AFM, NanoSurf Co. Ltd., Lausanne, Switzerland). For the morphological characterization of the film on glass, a Hitachi Tabletop Microscope TM3030 (Hitachi High-Tech, HITACHI, Toyo, Japan) was used at 5 kV. Due to the characteristics of the equipment, it was not necessary to coat the sample before being analyzed. The hybrid films and pristine porphine on glass were subjected to X-ray diffraction analysis using the θ–2θ technique, in a Rigaku Miniflex 600 diffractometer (Rigau Corporation, Tokyo, Japan), Cu Kα (λ = 1.5406 Å), at 40 kV, 20 mA. The absorbance and transmittance were obtained with the glass substrate in order to get the optical band gap, using an ultraviolet–visible (UV–vis) spectrophotometer 300 Unicam (Thermo Fisher Scientific Inc., Waltham, MA, USA), in a wavelength range from 200 to 1100 nm. The FTO substrates were used for measuring the current-voltage (I-V) characteristics. A Keithley 4200-SCS-PK1 auto-ranging picoammeter (Tektronix Inc., Beaverton, OR, USA) was used with the collinear four-point probe method. The samples were illuminated with commercial LEDs of wavelengths between 400 and 700 nm (from red color to near-ultraviolet), using a lighting controller circuit from Next Robotix (Comercializadora K Mox, S.A. de C.V., Mexico City, Mexico). For these LEDs, typical operating voltages varied between 1.8 and 2.8 V, with a typical operating current of 18 mA.

## 3. Results and Discussion

### 3.1. Structural and Morphological Characterization of Hybrid Films

The IR spectroscopy was performed with the purpose to identify the main bonds and chemical stability of CuP and polymers after the deposit of the hybrid films. FTIR spectra of films are illustrated in Figure 2, and the observed IR bands and their assignments are recorded in Table 1, which are in good agreement with the literature [14,23,24,25,26,27,28,29,30]. The presence of C–H groups was indicated by the bands at 2845 cm^−1^ (CH_3_ asymmetric stretching), at 2925 ± 3 cm^−1^ (symmetrical CH_2_), and 1441 ± 4, 1379 ± 7, 703 ± 1 cm^−1^ (asymmetrical CH_2_) [31,32,33,34], whereas the presence of C-C was indicated by the band at 1462 ± 2 cm^−1^ [14,23,24,25,26,27,28,29,30,34]. Additionally, the presence of a pyrrole C-N bond as indicated by the band at 1021 ± 2 cm^−1^ [14,23,24,25,26,27,28,29,30,34], and the bands at 1066 and 844 ± 2 cm^−1^ are related to N-H vibration [14,23,24,25,26,27,28,29,30,31,32,33]. On the other hand, the presence of PMMA in the PMMA:CuP film is verified by the signals at 1739, 1634, and 1134 cm^−1^ and the PS signals in the PS:CuP film are found at 3026, 2922, 1606, and 757 cm^−1^, and are shown with their respective assignments in Table 1. Accordingly, the presence of all these bands confirms the incorporation of CuP in the polymeric matrix, and this also suggests that the CuP is structurally unchanged during the preparation of hybrid films. However, it is necessary to analyze their topography, to determine if the CuP was adequately distributed in the polymeric matrix.

In order to know the surface topography and roughness of the PMMA-CuP and PS:CuP films, they were analyzed by AFM and Figure 3 shows the images at 5 × 5 μm. The hybrid films exhibit a very uniform topography with some ridges, probably related to preferential directions of growth. However, in both films, it would appear that the porphin is completely embedded in the polymeric matrix. Regarding the root mean square (RMS) roughness, the film with PMMA is less rough than the film with the PS matrix, the values obtained are 2.59 and 3.66 nm, respectively. The low roughness and adequate homogeneity of hybrid films make them susceptible to being used as components in optoelectronic devices; however, it is necessary to complement the morphological characterization by SEM.

In order to complement the surface morphology of the PMMA-CuP and PS:CuP films, they were analyzed by SEM and Figure 4 shows the images at 800×. It can be observed that both images exhibit the same continuous and homogenous surface, with minor imperfections due to the manufacturing process. This implies that the CuP is homogeneously distributed within the polymeric matrix in both films

The structure of the hybrid films and the pristine porphine was studied by X-ray diffraction and the diffraction patterns are given in Figure 5. In the PMMA:CuP film (Figure 5a), the XRD pattern exhibits a broad diffused peak at approximately 17°–34° without detectable sharp Bragg peaks that can be related to a crystalline phase. This implies that the film is essentially amorphous and the porphine has poor crystallinity in the film [35]. The PS:CuP film (Figure 5a), in addition to the broad diffused peak at approximately 14°–35°, presents sharp Bragg peaks at approximately 2θ = 7.11°, 7.89°, and 19° indicating some degree of π–π stacking of the CuP molecules in certain regions of the hybrid film, which suggest that the PS:CuP film has a crystalline structure [35]. Apparently, there is a small nucleation and crystallization of the CuP with a preferential direction of growth [35,36,37]. The greater crystallinity in the PS:CuP hybrid film could be reflected in better optical and electrical properties; however, it is important to consider that the porphine is embedded in an amorphous polymeric matrix; therefore, the predominant structure of the hybrid film is amorphous. On the other hand, Figure 5b shows the diffraction pattern for the CuP pristine film. It can be observed that porphine presents a sharp Bragg peak at approximately 2θ = 8°, related to a crystalline phase. It is important to consider that this film was deposited by evaporation under a high vacuum since it was not possible to manufacture it by the simple solution method used for the preparation of hybrid films. The use of PMMA and PS matrices facilitates the solution preparation of hybrid films.

### 3.2. Evaluation of Optical Parameters in Hybrid Films

In order to study the optical properties of the PMMA:CuP and PS:CuP films, UV–vis spectroscopy was carried out, and it is important to consider that these spectra are dominated by porphine π–π* transitions. The spectral distribution of transmittance (%T) and absorbance (A) as a function of the wavelength for the polymer:CuP films are shown in Figure 6. In Figure 6a it is shown that at a longer wavelength greater than 650 nm, the hybrid films become nearly transparent as there is no light being absorbed, while in the ultraviolet range, sharp band edge absorption is observed for the two films. The presence of such edges in these films makes them potential candidates for optical filter components. Although the transmittance obtained in the hybrid films reaches a maximum of 68% for PS:CuP and 84% for PMMA:CuP, this value is lower than that reported in the literature for other metallic porphine films, such as porphine iron (III) chloride (FeTPPCl) [11], manganese (III) chloride tetraphenyl porphine (MnTPPCl) [14], for platinum octaethylporphyrin PtOEP [36], and for cobalt porphine CoMTTPP [38] presenting %T > 90%. However, the spectral distribution of transmittance is similar to the previous examples, and above all, the PS:CuP film could be used (at wavelengths greater than 650 nm), as a transparent anode in photovoltaic devices.

Concerning the absorbance presented in Figure 6b, the spectra of PS:CuP shows three bands: an intense band at 401 nm (Soret-band), and a doublet at 526 nm (β-band) and 562 nm (∝-band) that is related to Q-bands. For PMMA:CuP the spectra show the three bands at 398 (Soret-band), 539 (β-band), and 586 nm (∝-band). The peaks of Soret and Q-bands are interpreted as the excitation between bonding and antibonding molecular orbital in the porphine, in terms of π–π* [14] and the results obtained for the PMMA:CuP and PS:CuP films coincide with those obtained for other structures with porphines [8,37]. The absorption capacity of the CuP makes hybrid films susceptible to participate in photoinduced charge transfer processes in optoelectronic and photovoltaic devices.

In organic semiconductors, it is usual to study the type of optical transitions in the material, as well as their value of the optical band gap. The transitions in the CuP are promoted from the valence band “π-band” to the conduction band “π*-band”, and these orbitals are separated by the band gap. There is an expression that relates to the absorption coefficient (α):α = ln(T/d)(1)
to the optical band gap (E_g_^opt^), and it is the Tauc’s relation [39,40,41]:(αhν)^r^ = G(hν − E_g_^opt^),(2)
where T is the transmittance, and d is the thickness of the film of 12 μm for PMMA:CuP and 78 μm for PS:CuP. The thickness was measured with the AFM, using the silicon cantilever, in an image size of 5 × 5 μm, with measurements at 256 points per line. Regarding Equation (2), h is the Planck constant, ν is the frequency obtained from ν = c/λ, λ is the wavelength, c is the speed of light, G is a parameter that depends on the transition probability, and r is a number which characterizes the type of electronic transitions, and where r = 2 for indirectly allowed transitions, corresponding to semiconductors films with a mainly amorphous structure. Moreover, the indirect allowed transition was observed in several metalloporphines such as iron [11], manganese [14], platinum [36], cobalt [38], and nickel [42]. The indirect band gap for the PMMA:CuP and PS:CuP films are evaluated by extrapolating the linear portion of the plot to (αhν)^1/2^ = 0. The obtained values are shown in Figure 7a,b, the first energy value is the onset band gap, E_g_^onset^ (1.89 and 2.07 eV for PMMA:CuP and PS:CuP, respectively), and corresponds to the formation of a bound electron–hole pair, or onset band gap [36,43]. The second energy (2.16 and 2.86 eV for PMMA:CuP and PS:CuP, respectively) is the optical band gap, E_g_^opt^, and corresponds to the energy gap between the valence band and the conduction band [11,43]. These results are important not only because they are in the same order of magnitude as those obtained for other porphine and their derivatives [36,37,38,39,40,41,42], but also because the PMMA or PS polymeric matrix does not increase the porphine band gap in hybrid films. One of the drawbacks in hybrid films, due to the presence of the polymeric matrix, is that it can reduce charge transport due to the interfaces that occur between the polymer and the porphine. In this case, E_g_^opt^ and the E_g_^onset^ are not affected, although surely in the films, there are a significant number of traps and defects that can be indirectly evaluated through the Urbach energy (E_U_). The E_U_ corresponds to the width of the band tail, which is related to localized states within the energy gap, possibly caused by structural defects [44]. The E_U_ can be used to determine the defects in the bandgap and can be determined according to the expression [44]:(3)$$\alpha =A_{a}exp\left(\frac{hv}{E_{U}}\right)$$

In addition to the parameters defined above, A_a_ is a constant of the material that conforms to α at the energy gap. The values of the E_U_ were determined from the reciprocal of the slope from this linear relation and have been recorded in Figure 7c,d. It is important to consider that the value of E_U_ is zero in a perfect semiconductor [45] and in this study, E_U_ of 0.70 and 0.25 eV have been obtained for the PMMA:CuP and PS:CuP films, respectively. The above results are indicative of the higher number of defects and traps in the film with the PMMA matrix, which is evident from the amorphous structure of PMMA:CuP shown in Figure 5a. On the other hand, it is remarkable how the film with the PS matrix presents a lower E_U_ than that obtained even in pristine porphine films such as the MnTPPCl one obtained by Alharbi et al. [14]. Apparently, the higher degree of porphine crystallinity observed on the XRD and presented in Figure 5a is responsible for the low E_U_; however, the matrix also influences the optical behavior of the films since the smallest optical and onset band gaps and the highest transparency are obtained in the film with a PMMA matrix compared to the PS matrix.

### 3.3. Evaluation of Electrical Behavior

To evaluate the electrical properties of the PMMA:CuP and PS:CuP films, the I–V characteristic curves were obtained for glass/FTO/PMMA:CuP/Ag and glass/FTO/PS:CuP/Ag simple devices under different incident light colors. It can be observed in Figure 8a, a change in the curve behavior for the natural light conditions of the glass/FTO/PMMA:CuP/Ag device. The rest of the curves for this device present the same behavior: the change in the type of radiation that falls on it does not affect the amount of current that flows through the device. At V < 0.84 V, the behavior is mainly ohmic and at V > 0.84 V there is a decrease in the amount of current carried, probably due to charge saturation at the polymer-CuP interfaces. As the voltage increases and with it, the concentration of charges in certain areas of the amorphous film, the flow of current becomes more difficult; however, at approximately 1 V, the charges flow again and the device continues with ohmic behavior. It is interesting to observe in Figure 8a how this saturation of charge carriers does not occur when changing the direction of the flow, reversing the anode and cathode. In this inverse sense, the electrical behavior remains ohmic throughout the analyzed voltage range, so it can be deduced that for the device with the PMMA:CuP film, the type of material that acts as the electrode is important, as well as the direction of flow of charges. Figure 8c shows the curves with the variation in the current density (J) with respect to the applied voltage, obtained for conditions of natural lighting and darkness. In these two curves, the difference in the flow of transported charge becomes evident: in the device under dark conditions, the J is an order of magnitude higher than in daylight conditions, suggesting that PMMA:CuP film can have photovoltaic properties and be used in the manufacturing of solar cells or photodiodes. However, it is possible to observe that the J in natural lighting conditions presents an ohmic behavior at V < 1.16 V and a Space Charge-Limited Current (SCLC) behavior at V > 1.16 V. This last mechanism is generated because, as previously mentioned, by increasing the voltage, a situation is reached in which the charge carriers do not move fast enough and accumulate in certain regions of the film. Under these conditions, the film is no longer homogeneous and the device enters the SCLC regime. The behavior of the glass/FTO/PMMA:CuP/Ag device shown in Figure 8c, under natural lighting conditions, is that of a resistor. Apparently the weak intra- and intermolecular interactions in the PMMA:CuP amorphous film hinder the flow of charge carriers.

On the other hand, the glass/FTO/PS:CuP/Ag device (Figure 8b) presents a similar behavior in all lighting conditions to which it was subjected, and the current it carries is an order of magnitude greater than in the device with a PMMA matrix. Furthermore, its behavior is not ambipolar either and is more related to that of a Schottky diode, with an exponential relationship between the applied voltage and the amount of electric current carried. In the curves of Figure 8b, a slight change in the slope of the curves is observed at approximately 0.9 V, although it is minimal, and based on the electrical behavior, it could be concluded that the PS:CuP film presents a greater potential to be used as a component in optoelectronic devices compared to the PMMA:CuP film. The greater order in the structure of this hybrid film offers less difficulty for the charge transporters to move through the energy bands and their mobility is greater. Finally, it is important to observe how the polymer used as a matrix exerts a marked influence on the optical and electrical properties of porphine films. While the PMMA matrix favors the transparency of the film and the low band gap values, the PS matrix increases the charge flow along the hybrid film and changes the behavior of the resistor generated by the PMMA matrix, to a Schottky diode-type device. However, the hybrid films are promising candidates for use as active layers in optoelectronic and photovoltaic devices.

## 4. Conclusions

Using a simple dispersion-polymerization technique, hybrid films were fabricated with copper porphine embedded in PMMA and PS polymeric matrices. The films were characterized structurally by IR and DRX and morphologically by AFM and SEM. The DRX analysis showed that the PMMA:CuP film is mainly amorphous and the PS:CuP has a crystalline structure. Although the optical properties of the films are dominated by the presence of CuP, the polymer used as a matrix can significantly modify them. The PMMA:CuP film has an optical band gap of 2.16 eV and an onset gap of 1.89 eV and the PS film has an optical band gap and onset gap of 2.86 eV and 2.07 eV, respectively. The electrical behavior in films is also affected by the type of polymer: the largest amount of current carried is approximately 0.01 A for the PS:CuP film, and 0.0015 A for the PMMA:CuP film. Furthermore, the behavior of the device with this hybrid film is similar to that of a resistor, while the device with the PS:CuP film is more similar to that of a Schottky diode. However, thanks to the above properties, hybrid films are promising candidates for use as an active layer in optoelectronic and photovoltaic devices.

## Figures and Tables

**Figure 1 polymers-15-03125-f001:**
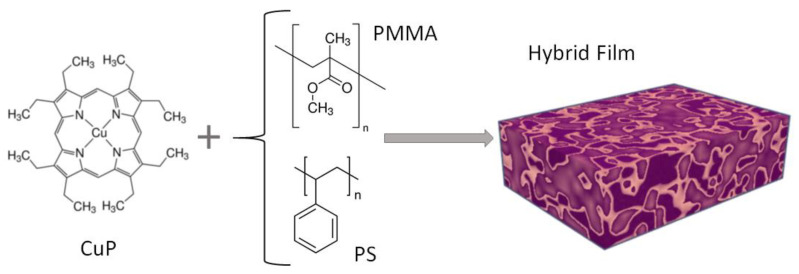
Structure of CuP, PMMA, and PS precursors.

**Figure 2 polymers-15-03125-f002:**
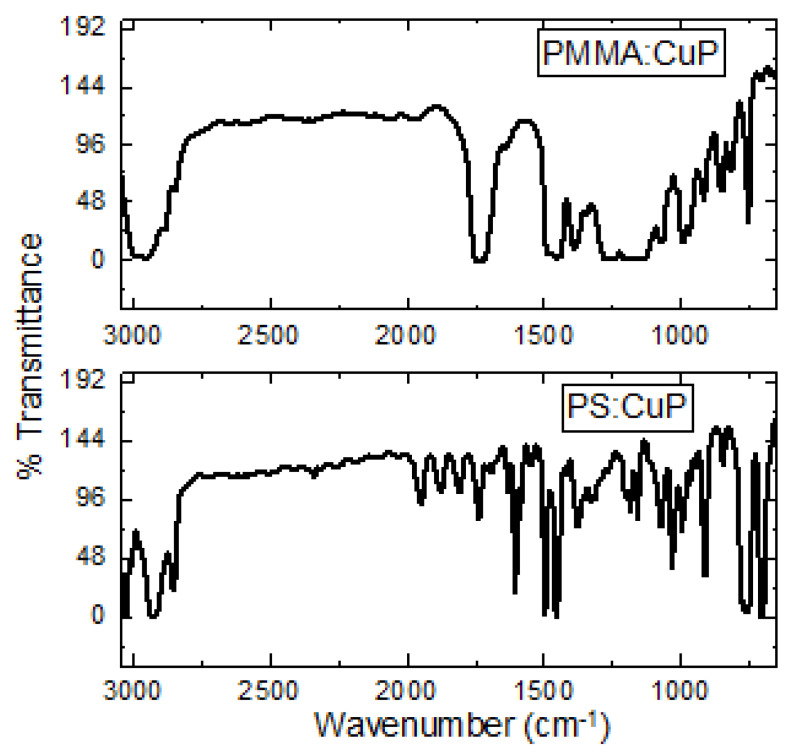
FTIR spectra of polymer-CuP films.

**Figure 3 polymers-15-03125-f003:**
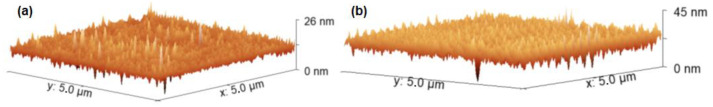
AFM images of the (**a**) PMMA:CuP and (**b**) PS:CuP films.

**Figure 4 polymers-15-03125-f004:**
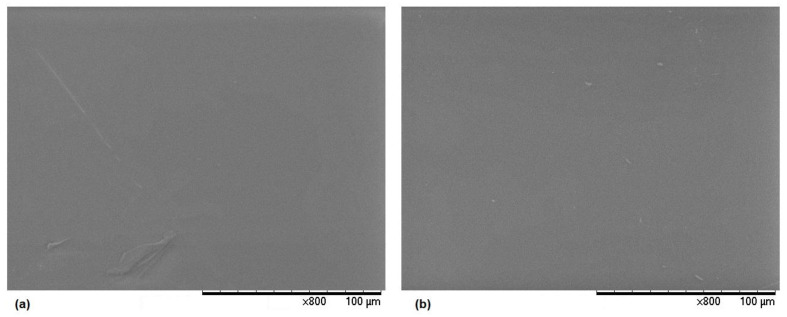
SEM images of the (**a**) PMMA:CuP and (**b**) PS:CuP films at 800×.

**Figure 5 polymers-15-03125-f005:**
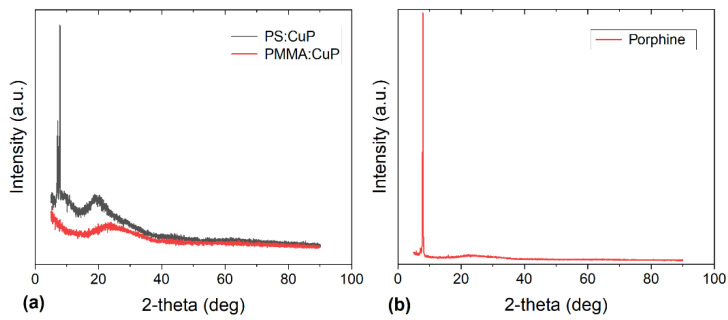
XRD patterns of the (**a**) polymer:CuP and (**b**) pristine CuP films.

**Figure 6 polymers-15-03125-f006:**
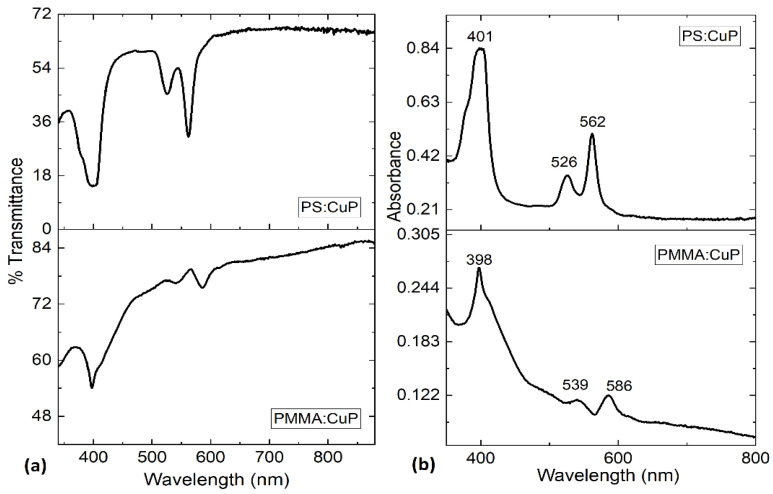
(**a**) %Transmittance and (**b**) absorbance spectra of polymer:CuP films.

**Figure 7 polymers-15-03125-f007:**
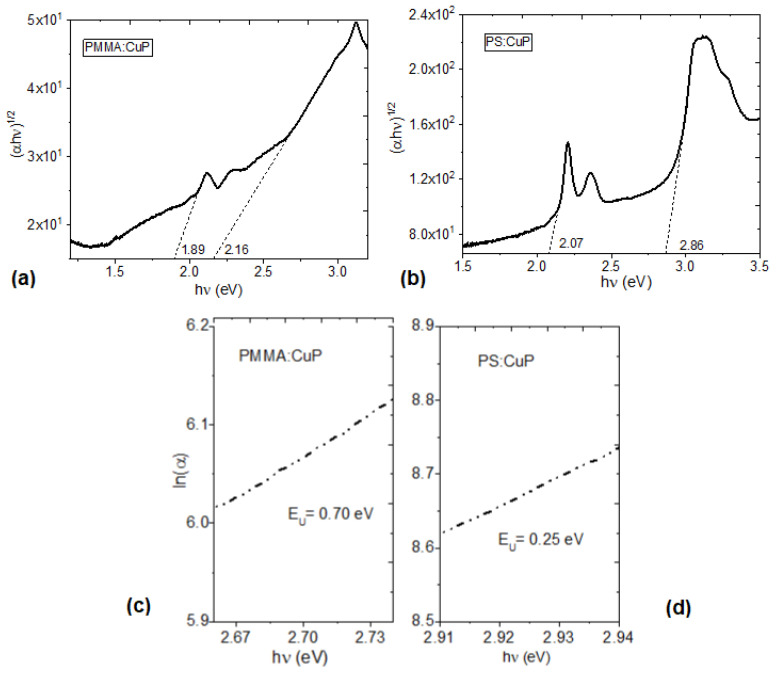
(**a**,**b**) Relation between (αhν)^1/2^ vs. hν energy and (**c**,**d**) relation between ln(α) vs. hν energy for the PMMA:CuP and PS:CuP films.

**Figure 8 polymers-15-03125-f008:**
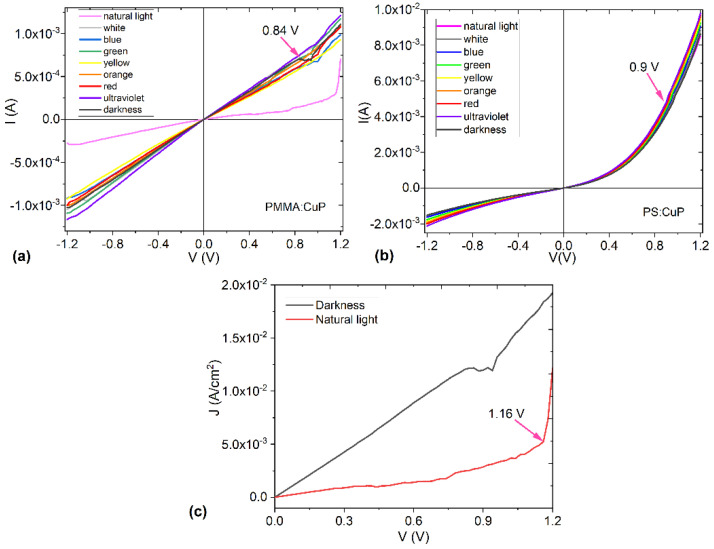
I-V characteristic curves of the (**a**) glass/FTO/PMMA:CuP/Ag and (**b**) glass/FTO/PS:CuP/Ag devices under different incident light color. (**c**) J–V characteristic curves of glass/FTO/PMMA:CuP/Ag under light and darkness conditions.

**Table 1 polymers-15-03125-t001:** Assignments of polymer-CuP films IR spectra.

PMMA:CuP (cm^−1^)	PS:CuP (cm^−1^)	Assignments
1445, 1386, 702	1437, 1372, 704	CuP: ν(C-H_2_)_asym_
2927	2924	CuP: ν(C-H_2_)_sym_
2845	2845	CuP: ν(C-H_3_)_asym_
1461	1464	CuP: ν(C-C)
1019	1023	CuP: ν(C-N)_pyrrole_
1066, 846	1066,843	CuP: ν(N-H)
1739		PMMA: ν(C=O)
1637		PMMA: ν(C=C)
1134		PMMA: ν(C-O)
	3026, 2922	PS: ν(C-H)
	1606	PS: ν(C=C)
	757	PS: ν(benzene)

## Data Availability

Data is contained within the article.

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
