# Peer review of "Influence of the Polymeric Matrix on the Optical and Electrical Properties of Copper Porphine-Based Semiconductor Hybrid Films"

_polymers, 2023, doi:10.3390/polym15143125_

Round 1
Reviewer 1 Report
In this paper, the authors studied the electrical and optical behavior of semiconductor hybrid films fabricate from CuP embedded in polymethylmethacrylate (PMMA), and polystyrene (PS), respectively, to afford PMMA:CuP and PS:CuP hybrids. The systematical comparison study between two kinds of hybrids in terms of structures, properties and electrical performances, which showed that PS:CuP film hybrid had higher current (0.01 A). Overall, this study was well designed and its novelty is sufficient. The manuscript has also been well written. Its content meets well the scope of Polymers. I would like recommend it for publication in Polymers after minor revision:
1) The key words: PMMA and PS are suggested to be listed as key words.
2) Line 43: The authors stated that “…chemical properties, in contrast to small molecule materials that must be thermally evaporated [2]…..” It’s not true. In fact, numerous solution processable small molecules used in organic electronics have been developed and reported, such as ITIC, Y6 etc used on organic solar cells.
3) Figure 4: The authors stated that PS:CuP film have a crystalline structure. The XRD of pristine CuP should be provided and incorporated in Figure 4 for better comparison,
The language has no major problems.
Author Response
Comments and Suggestions for Authors
In this paper, the authors studied the electrical and optical behavior of semiconductor hybrid films fabricate from CuP embedded in polymethylmethacrylate (PMMA), and polystyrene (PS), respectively, to afford PMMA:CuP and PS:CuP hybrids. The systematical comparison study between two kinds of hybrids in terms of structures, properties and electrical performances, which showed that PS:CuP film hybrid had higher current (0.01 A). Overall, this study was well designed and its novelty is sufficient. The manuscript has also been well written. Its content meets well the scope of Polymers. I would like recommend it for publication in Polymers after minor revision:
1) The key words: PMMA and PS are suggested to be listed as key words.
Answer. Polymethylmethacrylate and Polystyrene keywords were included.
2) Line 43: The authors stated that “…chemical properties, in contrast to small molecule materials that must be thermally evaporated [2]…..” It’s not true. In fact, numerous solution processable small molecules used in organic electronics have been developed and reported, such as ITIC, Y6 etc used on organic solar cells.
Answer. Thank you for the correction. The Introduction in the manuscript was revised, corrected and an appropriate bibliographical reference was included.
3) Figure 4: The authors stated that PS:CuP film have a crystalline structure. The XRD of pristine CuP should be provided and incorporated in Figure 4 for better comparison,
Answer. The XRD of pristine CuP was included in the text as Figure 5b. The presence of a crystalline structure can be observed.

Reviewer 2 Report
The work on Cu porphine complexes blended with PMMA and PS appear as a new contribution to the area of organic electronics. The work is complete, well designed, and detailed in characterization. I may only suggest the authors clarify what type of PS they use; I assume it was amorphous atactic. Also, DSC studies might shed light on the crystallization of Cu blends, but not necessary since this work is heavily focused on optics.
Some minor proofreading required.
Author Response
Comments and Suggestions for Authors
The work on Cu porphine complexes blended with PMMA and PS appear as a new contribution to the area of organic electronics. The work is complete, well designed, and detailed in characterization. I may only suggest the authors clarify what type of PS they use; I assume it was amorphous atactic. Also, DSC studies might shed light on the crystallization of Cu blends, but not necessary since this work is heavily focused on optics.
Answer. Thanks for the observation. The polystyrene used in this work is amorphous and atactic. This information was included in the experimental section and the XRD discussion section of the manuscript.
Comments on the Quality of English Language
Some minor proofreading required.
Answer. The English language was carefully checked and corrected in the manuscript.

Reviewer 3 Report
I enjoy reading the paper. However, the authors should address the following comments before the paper is accepted by Polymers.
1. Why did the author discuss a lot about YD2-oC8 in Introduction? Is it related to the work in this paper?
2. At line 83, the author introduced PMMA and PS after introducing conductive polymers. Are they conductive polymers? If not, the author needs to change the structure of Introduction.
3. I don’t think the author finished the last sentence in Introduction.
4. The AFM tip and mode (tapping or contact) details are needed in Materials and Methods.
5. The caption for Figure 1 is wrong.
6. Is there any coating performed on the sample surface when doing SEM imaging?
7. At line 230, how did the author measure the thickness of the films?
8. I suggest the author labeling the critical value 0.8 V in Figure 7a 0.9 V in Figure 7b and 1.1 V in Figure 7c.
I suggest a minor revision.
Author Response
Comments and Suggestions for Authors
I enjoy reading the paper. However, the authors should address the following comments before the paper is accepted by Polymers.
- Why did the author discuss a lot about YD2-oC8 in Introduction? Is it related to the work in this paper?
Answer. Thanks for the remark. The information about the solar cell composed of YD2-o-C8 has been summarized. Although there is a relationship with the optical behavior of YD2-o-C8, there was excessive information in the Introduction that was not necessary, and it has been eliminated.
- At line 83, the author introduced PMMA and PS after introducing conductive polymers. Are they conductive polymers? If not, the author needs to change the structure of Introduction.
Answer. The structure of the Introduction was revised and modified.
- I don’t think the author finished the last sentence in Introduction.
Answer. Thanks for your observation. The sentence was revised and completed.
- The AFM tip and mode (tapping or contact) details are needed in Materials and Methods.
Answer. The tip and mode details were included in the experimental section.
- The caption for Figure 1 is wrong.
Answer. Figure 1 was separated into two figures, currently labeled as Figure 1 and Figure 2. Figure 1 shows the structures of the CuP, PMMA and PS precursors. Figure 2 presents the IR spectra of the hybrid films.
- Is there any coating performed on the sample surface when doing SEM imaging?
Answer. No, the samples were placed on the sample holder and inserted into the microscope's vacuum chamber without any coating. To avoid ambiguities, a statement was added to the text indicating the above.
- At line 230, how did the author measure the thickness of the films?
Answer. The thickness was measured with an atomic force microscope. The details of the measurement were supplemented in the manuscript.
- I suggest the author labeling the critical value 0.8 V in Figure 7a 0.9 V in Figure 7b and 1.1 V in Figure 7c.
Answer. Critical values have been included in the figures.
Comments on the Quality of English Language
I suggest a minor revision.
Answer. The English language was carefully checked and corrected in the manuscript.